# Inclusion of Metabolic Tumor Volume in Prognostic Models of Bone and Soft Tissue Sarcoma Increases the Prognostic Value

**DOI:** 10.3390/cancers15030816

**Published:** 2023-01-28

**Authors:** Mette Abildgaard Pedersen, Thomas Baad-Hansen, Lars C. Gormsen, Steen Bærentzen, Birgitte Sandfeld-Paulsen, Ninna Aggerholm-Pedersen, Mikkel Holm Vendelbo

**Affiliations:** 1Department of Nuclear Medicine & PET Centre, Aarhus University Hospital, 8200 Aarhus N, Denmark; 2Institute of Biomedicine, Aarhus University, 8200 Aarhus N, Denmark; 3Steno Diabetes Centre Aarhus, Aarhus University Hospital, 8200 Aarhus N, Denmark; 4Department of Orthopedics, Aarhus University Hospital, 8200 Aarhus N, Denmark; 5Department of Pathology, Aarhus University Hospital, 8200 Aarhus N, Denmark; 6Department of Clinical Biochemistry, Viborg Regional Hospital, 8800 Viborg, Denmark; 7Department of Clinical Medicine, Aarhus University, 8200 Aarhus N, Denmark; 8Department of Oncology, Aarhus University Hospital, 8200 Aarhus N, Denmark

**Keywords:** sarcoma, 18F-FDG PET/CT, prognostic model, biomarker, survival

## Abstract

**Simple Summary:**

Sarcoma is a rare cancer originating in soft tissue or bone. Prognostic models are used to modify therapy and improve survival. The present study aimed to evaluate if the combination of PET parameters and circulating biomarkers can improve the prognostic accuracy. When PET parameters were added to the existing models, the prognostic value increased in all models. A new prognostic model (SBSpib), including the biomarkers albumin, lymphocytes, and one PET parameter, metabolic tumor volume, was developed. SBSpib separates patients into four different groups, where the chance of survival increases with decreasing scores. Overall, combining PET parameters and circulating biomarkers improves the prognostic value. However, SBSpib must be validated before implementation.

**Abstract:**

Sarcomas are rare and have a high mortality rate. Further prognostic classification, with readily available parameters, is warranted, and several studies have examined circulating biomarkers and PET parameters separately. This single-site, retrospective study aimed to examine the prognostic values of several scoring systems in combination with PET parameters. We included 148 patients with sarcoma, who were treated and scanned at Aarhus University Hospital from 1 January 2016 to 31 December 2019. The Akaike information criterion and Harrell’s concordance index were used to evaluate whether the PET parameters added prognostic information to existing prognostic models using circulating biomarkers. Of the PET parameters, metabolic tumor volume (MTV) performed best, and when combined with the existing prognostic models, the prognostic value improved in all models. Backward stepwise selection was used to create a new model, SBSpib, which included albumin, lymphocytes, and one PET parameter, MTV. It has scores ranging from zero to three and increasing hazard ratios; HR = 4.83 (1.02–22.75) for group one, HR = 7.40 (1.6–33.42) for group two, and HR = 17.32 (3.45–86.93) for group three. Consequently, implementing PET parameters in prognostic models improved the prognostic value. SBSpib is a new prognostic model that includes both circulating biomarkers and PET parameters; however, validation in another sarcoma cohort is warranted.

## 1. Introduction

Sarcomas are rare and represent a heterogenic group of tumor entities. They are divided into soft-tissue sarcomas (STS) and bone sarcomas (BS) [1] and can be further divided into several histological subtypes. Sarcomas represent fewer than 1% of all new cancers. The five-year survival rate decreases significantly to 15–39% if there is metastatic disease [2,3,4,5].

Prognostic indicators are used to modify therapy and improve survival. Conventional prognostic markers include presence of metastases, resectability, surgical margins, size, location, tumor grade, and tumor burden [6,7]. With improved prediction, patients at low risk of recurrence may be spared potentially unnecessary treatment, whereas patients at high risk can be considered for intensified follow-up and/or adjuvant therapy. It is warranted to improve the prognostic classification, preferably using readily available parameters. Several studies have been performed to identify new valuable presurgical prognostic factors, but no consensus has been found. 

Visual inspection of 18F-FDG PET/CT scans can be used in the clinical staging of sarcomas. Several quantifiable parameters can be obtained from the 18F-FDG PET images and may be used as prognostic indicators. One is the standardized uptake value (SUV), a semiquantitative measure of tissue radiotracer uptake and an approximation of tumor metabolism. Pre-treatment tumor maximum SUV (SUVmax) has in some studies been shown to predict sarcoma grade and survival [8,9,10,11,12,13], although other groups have failed to reproduce this [14,15,16]. SUVmax reflects the radioactivity in a single voxel and is prone to artefacts from both image noise and scanner resolution. It reflects neither tumor size nor disease burden; however, other PET metrics may overcome this limitation. Metabolic tumor volume (MTV) is defined as the volume of tumor tissue with an SUV above a minimum threshold, often 2.5 g/mL. Although some studies using MTV to predict outcome in sarcomas have been promising, results are conflicting. Thus, it has been reported that MTV can predict survival [10,13,16,17] but also that MTV has no predictive value in risk stratifying sarcoma patients [18]. The MTV metric may be expanded by multiplying it with the mean SUV, yielding the metric total lesion glycolysis (TLG). TLG is the most accurate measurement of any lesion’s overall metabolic activity. TLG has been demonstrated to predict overall survival in patients with high-grade bone and soft tissue sarcomas [18]. However, results using this PET metric have also been conflicting [8,10,11,13,16,17,19,20,21].

Circulating biomarkers also have prognostic potential. In various studies, elevated c-reactive protein (CRP) is associated with inferior survival in patients with sarcomas [22,23,24,25,26,27,28,29]. There have been conflicting results on the predictive value of lactate dehydrogenase (LDH) [30,31] and hemoglobin [22,26,27,29]. Several other biomarkers have been examined, and studies have suggested that plasma fibrinogen [32], erythrocyte sedimentation rate (ESR) [25], and albumin [22] could be used as prognostic biomarkers. 

Attempts to include several biomarkers in scoring systems have been tried; the ‘high-sensitivity modified Glasgow prognostic score’ (HS-mGPS), a combination of hypoalbuminemia and elevated CRP, was an independent predictor for survival in soft tissue sarcoma [33]. Choi et al. found a model (Choi model) where the elevation of two or three of the biomarkers CRP, ESR, and neutrophil/lymphocyte ratio (NLR) were significant predictors of survival [25]. The Aarhus composite biomarker score (ACBS), including CRP, albumin, haemoglobin, neutrophil, and leucocyte counts, was a prognostic index for localized sarcoma [34,35]. The modified ACBS (ACBSm) was prognostic for metastatic sarcoma [36].

So far, no consensus on which staging classifications and prognostic indicators to use in a clinical setting has been found [7]. This study aimed to examine the prognostic values of several scoring systems in combination with PET parameters. To the best of our knowledge, this is the first study to examine the prognostic value of combining circulating biomarkers and PET/CT-based parameters.

## 2. Materials and Methods

### 2.1. Study Population and Design

In Denmark, all patients with sarcoma are treated at one of two sarcoma centers. This study was based at one of the two: Aarhus Sarcoma Centre. It was a single-site, retrospective study that included patients with soft tissue and primary bone sarcoma, all of whom were treated and scanned at Aarhus University Hospital from 1 January 2016 to 31 December 2019. The study protocol was approved by the Central Denmark Region and the institutional review board (no. 1-45-70-29-20), and informed consent was waived.

The patients were identified from a scan log coded as PET/CT for sarcoma and were included in the study according to the following criteria: (1) newly diagnosed sarcoma; (2) pre-therapeutic 18F-FDG PET/CT; (3) one or more baseline biomarkers evaluated; and (4) age over 18 years. The nomenclature <4 is used for groups consisting of three or fewer patients to comply with the General Data Protection Regulation (GDPR).

### 2.2. Biomarkers

The baseline plasma levels of albumin, haemoglobin, LDH, alanine aminotransferase (ALT), ESR, CRP, neutrophil count, lymphocyte count, and sodium were collected if they were taken no more than 30 days prior to or 14 days after the 18F-FDG PET/CT scan. The biomarkers were divided into normal or low/high levels. Normal levels were defined as: albumin > 37 g/L, haemoglobin > 7.3 mmol/L in women and >8.3 in men, LDH < 205 U/I, ALT < 45 U/I for women and <70 U/I for men, ESR < 20 mm/h, CRP < 8 mmol/L, neutrophils < 7 × 10^9^/L, lymphocytes > 1.3 × 10^9^/L, and sodium > 137 mmol/L.

### 2.3. 18F-FDG PET/CT

The 18F-FDG PET/CT was a pre-therapeutic scan performed as part of the diagnostic process. This is standard procedure in Denmark, and all newly diagnosed grade II-III sarcoma patients undergo a PET/CT scan. The scan was carried out in accordance with local protocols and manufacturer guidelines. Whole-body examinations were performed from the vertex level of the skull or orbits through to the feet. Patients had fasted for a minimum of 6 h before intravenous administration of 18F-FDG (196–604 MBq). 

SUV was defined as the concentration of 18F-FDG corrected for injected dose, the patient’s body weight, and radioactive decay at the time of scanning. SUVmax was measured on attenuation-corrected 18F-FDG PET/CT images with a volume of interest (VOI) covering the entire tumor. MTV2.5 and TLG2.5 were measured from attenuation-corrected 18F-FDG PET/CT images using an automated contouring program (ACCURATE) [37] and SUV threshold of 2.5 g/mL; this threshold has previously been used by others [10,16,17]. A VOI is visualised in Figure 1. Each VOI was checked and approved manually by MAP. The total MTV2.5 was calculated automatically by summing the volumes of voxels with a threshold of SUV 2.5 g/mL in the tumor VOI. TLG2.5 is the product of SUVmean and MTV2.5.

### 2.4. Current Prognostic Scores

When examining the literature, we found four existing prognostic scores for sarcoma patients: HS-mGPS, Choi model, ACBS, and ACBSm. The HS-mGPS [33] included albumin and CRP. Patients were allocated a score of two if they had hypoalbuminemia and elevated CRP. If they only had elevated CRP, they were allocated a score of one. The remaining had a score of zero. The Choi model [25] consisted of ESR, CRP, and NLR. The score was zero if none or one of the biomarkers were elevated and one if two or three were elevated. ACBS [34,35] included five biomarkers: albumin, haemoglobin, CRP, neutrophils, and lymphocytes. The score was zero if all biomarkers were normal, one if one of the biomarkers was abnormal, and two if more than one was abnormal. ACBSm [36] included serum sodium in addition to ACBS biomarkers and was divided into four groups. 

### 2.5. New Prognostic Scores

In the present study, we examined the value of combining biomarkers and PET parameters in new scores by adding MTV2.5 to each score. All scores were expanded with an additional group when MTV2.5 was added. The following new scores were formed (Appendix A). MTV_HS-mGPS: The score was zero if CRP was normal and MTV2.5 was <56 mL; one if CRP was elevated or MTV2.5 was >56 mL; two if CRP was elevated, albumin normal and MTV2.5 > 56 mL; and three if both biomarkers were abnormal and MTV2.5 > 56 mL. MTV_Choi: The score was zero if none of the parameters were high/elevated and MTV2.5 < 56 mL; one if two or three parameters were high/elevated including MTV2.5 > 56 mL; and two if all three parameters were high/elevated and MTV2.5 > 56 mL. MTV_ACBS: The score was zero if all biomarkers were normal and MTV2.5 < 56 mL; one if one of the biomarkers was abnormal or MTV2.5 was >56 mL; two if two factors were abnormal (incl. MTV2.5 > 56 mL); and three if three or more were abnormal. MTV_ACBSm: The score was zero if all biomarkers were normal and MTV2.5 < 56 mL; one if one of the biomarkers was abnormal or MTV2.5 was >56 mL; two if two factors were abnormal (incl. MTV2.5 > 56 mL); three if three were abnormal; and four if four or more factors were abnormal. Finally, we developed a new scoring model based on our data, as outlined in the statistical analysis section.

### 2.6. Statistical Analysis

Descriptive statistics were used to calculate the baseline characteristics. Continuous data are presented as mean (range) in case of normality and as median (range) when log transformation results in normal distributions. QQ-plots checked the normality of the data. Categorical biomarkers were categorised using thresholds equivalent to the ones used in a clinical setting at our institution and presented as the number of patients, *n*(%), in each group. Biomarkers were investigated separately, in a Cox proportional univariate analysis, and when statistically significant, they were also investigated as part of a multivariate analysis.

Overall survival (OS) was defined as the time from diagnosis until death from any cause. The prognostic value of the PET parameters was determined by analyzing the areas under the receiver operating characteristic (ROC) curves, and cut-off values were determined via the Youden index [38]. Kaplan–Meier estimates, the log-rank test, and Cox proportional hazards were used to assess the survival functions’ equality across variables in the survival analyses. Relative risk was used to correlate tumor volume and metastatic disease.

To compare PET parameters, current prognostic scores, and the new combined scores, the Akaike information criterion (AIC) and the Harrell’s concordance index (C-index) were used. For this comparison, no missing values were allowed. Since MTV2.5 was available for all patients included, the number of patients in the analyses were dictated by the number of patients, with no missing values, in the current prognostic scores. Backward stepwise selection was used to create the new prognostic model, with 0.06 as the significance level for removal from the model and 0.05 as the significance level for addition to the model. The score was based on patients with localized disease exclusively, and biomarkers with more than 60% missing values were excluded before the backward stepwise selection; no missing values were allowed in the analysis. All analyses were carried out using STATA version 17.0 (StataCorp. 2021. Stata Statistical Software: Release 17. College Station, TX: StataCorp LLC) and GraphPad Prism version 9.3.1 (GraphPad Software, San Diego, CA, USA).

## 3. Results

### 3.1. Patients and Tumor Characteristics

From 1 January 2016 to 31 December 2019, 435 patients underwent an FDG PET/CT using the sarcoma scanning protocol at our institution. Of these, 151 patients were scanned pre-therapeutically and had a final diagnosis of sarcoma. Two scans did not cover the entire lesion and one could not be processed in ACCURATE, leaving a total of 148 patients to be included in the study.

Of the 148 included patients, 70 were women and 78 were men, with an average age of 61 years (range: 18–92 years). The mean follow-up period was 2.8 years (range: 0.1–6.2 years). The tumors comprised 109 (74%) STS and 39 (26%) BS; for further detail on histological subtypes, see Table 1. The median SUVmax was 13.53 g/mL (range: 1.68–62.29 g/mL), the median MTV2.5 was 73.89 mL (range: 0–2264.06 mL), and the median TLG2.5 was 351.45 g (range: 0–15,364.82 g). With a low R squared (R^2^ = 0.09), MTV2.5 positively correlated with SUVmax (*p* < 0.01, R^2^ = 0.09) (Figure 2). The primary tumor sites were the lower extremity (*n* = 61), pelvis (*n* = 21), upper extremity (*n* = 17), thorax (*n* = 12), breast (*n* = 8), head (*n* = 6), abdomen (*n* = 8), shoulder (*n* = 6), buttock (*n* = 6), back (*n* < 4), and other sites (*n* < 4). In 118 (79%) patients, the tumors were localized at diagnosis, while 31 (21%) had disseminated disease. Table 1 shows their baseline demographics.

### 3.2. Survival

A total of 71 (48%) patients died during follow-up; of these, 27 (38%) had metastatic disease at diagnosis, while 44 (62%) had localized disease. Among the patients who had localized disease and died during follow-up, 4 (9%) were inoperable at diagnosis. There was no difference in OS when comparing BS to STS, *p* = 0.57 (Appendix A). Therefore, they were analyzed as one cohort. Comparing survival in patients with localized disease at diagnosis to survival in patients with the metastatic disease showed a highly significant difference, *p* < 0.001 (Appendix A).

### 3.3. Baseline Circulating Biomarkers and Survival

Low albumin, low haemoglobin, high LDH, high CRP, high neutrophils, low lymphocytes, and low sodium were significantly correlated to inferior OS in univariate analyses. However, only low albumin, with HR = 4.18 (95% CI: 1.79–9.76), and high LDH, with HR = 3.80 (95% CI: 1.77–8.17), stayed significantly correlated to OS in the multivariate analysis. When examined as continuous variables, albumin, haemoglobin, LDH, CRP, and neutrophils were significantly associated with OS. In the multivariate analysis, this was only the case for LDH and neutrophils, with HR = 1.01 (95% CI: 1.00–1.01) and HR = 1.09 (95% CI: 1.01–1.18), respectively, for every 1.0 increase.

In patients with localized disease, low albumin was present in 31 (35%), low haemoglobin in 21 (22%), high LDH in 7 (10%), high ALT in 1 (3%), high ESR in 20 (47%), high CRP in 34 (44%), high neutrophils in 14 (18%), low lymphocytes in 25 (32%), and low sodium in 9 (9%) patients. In this subgroup, high LDH, high neutrophils, and low lymphocytes were significant predictors of inferior OS based on univariate analyses. Though, in the multivariate analysis, only low lymphocytes remained a significant predictor of inferior survival, with HR = 2.83 (95% CI: 1.24–6.45). When examined as continuous variables, albumin, haemoglobin, ALT, and CRP were significantly associated with OS. In the multivariate analysis, none of the circulating biomarkers stayed correlated with OS.

### 3.4. PET Parameters, Survival, and Metastatic Disease

The ROC curve analyses of OS for SUVmax, MTV2.5, and TLG2.5 with the area under the curve (AUC) data are presented in Figure 3a. For SUVmax, AUC = 0.69 (95% CI: 0.60–0.77); for MTV2.5, AUC = 0.72 (95% CI: 0.64–0.80); and for TLG2.5, AUC = 0.72 (95% CI: 0.63–0.80), with no statistically significant difference (*p* = 0.76). The optimal cut-off values for the PET parameters were as follows: SUVmax: 16 g/mL, MTV2.5: 56 mL, and TLG2.5: 1056 g. Dividing the cohort according to the optimal cut-off values resulted in statistically significant differences in survival, *p* < 0.001 for all groups (Figure 3b–d), with the following mortality rates: SUVmax (low vs. high): 31% vs. 68%, MTV2.5 (low vs. high): 27% vs. 64%, and TLG2.5 (low vs. high): 35% vs. 77%. This corresponded to hazard ratios of SUVmax: HR = 3.16 (95% CI: 1.94–5.15), MTV2.5: HR = 3.66 (95% CI: 2.11–6.32) and TLG2.5: HR = 3.91 (95% CI: 2.44–6.27).

Furthermore, the risk of metastatic disease at diagnosis was highly dependent on PET parameters. For this analysis, SUVmax had a significantly lower AUC compared to MTV2.5 and TLG2.5 (*p* = 0.007), and it was excluded from the analysis. Among patients with a high MTV2.5, 29 out of 84 (35%) had metastatic disease, while only 2 out of 64 (3%) patients had metastatic disease in the low MTV2.5 group, RR = 11.05 (95% CI: 2.74–44.60). Metastases were present in 22/47 (47%) patients with a high TLG2.5, while it was the case for 9/101 (9%) patients in the low TLG2.5 group, RR = 5.25 (95% CI: 2.62–10.52). We looked closer into the cases with metastatic disease despite MTV2.5 <56 mL; one patient had large tumor masses with weak to moderate FDG uptake resulting in a small MTV (5.19 mL) that differed from the actual tumor volume (>56 mL) measured on CT, and the other patient had epithelioid sarcoma in three lymph nodes, but the primary tumor was never localized.

For patients with localized disease at diagnosis, ROC curve analyses of SUVmax, MTV2.5, and TLG2.5 with AUC data are presented in Figure 4a. For SUVmax, AUC = 0.63 (95% CI: 0.53–0.74); for MTV2.5, AUC = 0.64 (95% CI: 0.53–0.74); and for TLG2.5, AUC = 0.63 (95% CI: 0.53–0.74), with no difference observed (*p* = 0.95). For SUVmax and MTV2.5, the optimal cut-offs were the same as in the total cohort; SUVmax: 16 g/mL and MTV2.5: 56 mL, respectively. However, for TLG2.5, the optimal cut-off increased to 1078 g. When applying these cut-offs, the following mortality rates were found: SUVmax (low vs. high): 27% vs. 55%, MTV2.5 (low vs. high): 26% vs. 51%, and TLG2.5 (low vs. high): 30% vs. 64%. This corresponded to hazard ratios of SUVmax: 2.49 (95% CI: 1.38–4.52), MTV2.5: 2.50 (95% CI: 1.35–4.63), and TLG2.5: 2.82 (95% CI: 1.52–5.23) for high vs. low SUVmax, MTV2.5, and TLG2.5, respectively.

### 3.5. Relapse in Localized Disease

Of the 113 patients with operable localized disease at diagnosis, 52 (46%) experienced relapses during follow-up: local recurrence (*n* = 23), metastatic recurrences (*n* = 34), and both local and metastatic relapses (*n* = 5). During follow-up, 35 (67%) patients with relapse died, while five (8%) patients with no relapse died. Among the five patients with no relapse, one died of complications to treatment, while the remaining four died of other causes. 

The median time from diagnosis to first relapse was 274 days, and it ranged from 53 to 1813 days. For 24 patients, the relapse was located in the lungs. Other locations included multiple organs (*n* = 6), lymph nodes (*n* < 4), pleura (*n* < 4), finger (*n* < 4), and abdomen (*n* < 4). The most common relapsed sarcoma types were epithelial sarcoma, rhabdomyosarcoma, and synovial sarcoma.

All PET parameters could predict relapse in localized disease: SUVmax: HR = 1.99 (95% CI: 1.15–3.44), MTV2.5: HR = 1.90 (95% CI: 1.10–3.29), and TLG25: HR = 2.08 (95% CI: 1.12–3.85).

### 3.6. Current Predictive Models

In the entire cohort, the highest scores of HS-mGPS, ACBS, and ACBSm were significant predictors of OS, whereas the Choi model could not predict OS (Appendix A). The pattern was the same for patients with localized disease at diagnosis, with the highest scores in HS-mGPS, ACBS, and ACBSm being significant predictors of OS, while the Choi model did not reach statistical significance.

We examined if the current models could predict future relapse in patients with localized disease at the time of diagnosis. We found that only the highest scores in HS-mGPS (HR = 2.64, 95% CI: 1.25–5.58, *p* = 0.01) and mACBS (HR = 2.78, 95% CI: 1.21–6.38, *p* = 0.02) were predictors of future relapse.

### 3.7. PET Parameters and Biomarkers Combined

PET parameters, current predictive models, and the new combined models were compared using AIC and C-index (Appendix A). Overall, MTV2.5 performed better than SUVmax and TLG2.5 according to AIC and C-index. We then combined MTV2.5 with the current models and found that the prognostic value improved for all models. This was evident for all sarcoma patients as a whole. For patients with localized disease, the overall picture was the same; however, AIC was unchanged for MTV_Choi, and the C-index dropped 0.1 for MTV_ACBSm (Table 2). 

The combination of PET parameters and the pre-existing models resulted in the following new predictions (Table 3). For MTV_HS-mGPS, we found that the two highest scores were now significant predictors of OS, with HR = 5.35 (95% CI: 1.89–15.12) and HR = 5.40 (95% CI: 2.13–13.66), respectively. The same pattern was evident for MTV_Choi, where HR for the highest score was statistically significant; HR = 4.76 (95% CI: 1.31–17.26). MTV_ACBS were divided into four groups, and the two highest scores had a significantly worse outcome than the reference group with HR = 5.53 (95% CI: 1.19–25.63) and HR = 10.40 (95% CI: 2.47–43.84), respectively, for OS. Finally, in MTV_ACBSm with five groups, the three highest scores had a significantly worse outcome. To visualise the predictive value of the updated models, we performed a calibration plot (Appendix A). For the predictive value in localized disease, see Table 3.

For predicting relapse in localized disease, the highest scores in MTV_HS-mGPS, MTV_ACBS, and MTV_mACBS were statistically significant, with HR = 4.01 (95% CI: 1.57–10.25, *p* < 0.01), HR = 3.18 (95% CI: 1.15–8.78, *p* = 0.03) and HR = 3.88 (95% CI: 1.36–11.08, *p* = 0.01), respectively. MTV_Choi could not predict relapse.

### 3.8. New Model

When creating the new prognostic model, we included only patients with localized disease and excluded ALT and SR from the analysis due to the high incidence of missing values (>60%). Backward stepwise selection was then used, starting with the full model, including circulating biomarkers and PET parameters. The final model included albumin, lymphocytes, and MTV2.5, with total scores from zero to three and increasing HR for each group (Table 4). 

Kaplan–Meier curves for the new model, Sarcoma Biomarker Score pet imaging and blood (SBSpib), in patients with localized disease are shown in Figure 5. The SPSpib model had an AUC = 0.75 for patients with localized disease. It significantly predicted relapse with HR = 5.26 (95% CI: 1.81–15.30) for group three. For groups one and two, the results did not reach significance, with HR = 1.35 (95% CI: 0.49–3.73) and HR = 2.15 (95% CI: 0.82–5.67), respectively.

## 4. Discussion

To the best of our knowledge, this is the first study to combine PET parameters and biomarkers in a predictive model. The main finding of this study is that adding PET parameters in the form of SUVmax, MTV2.5, or TLG2.5 to existing prognostic scores caused improvements in all of them. Overall, MTV2.5 outperformed SUVmax and TLG2.5 in the combined models. This suggests that a future prognostic model should contain MTV2.5 as one of the parameters. To make a suggestion, based on our data, backward stepwise selection was used to build a model that combined PET parameters and circulating biomarkers. This resulted in the SBSpib model with increasing HRs for each group. This model is based on a small data set consisting of 65 patients with localized disease and is therefore not readily implemented. It has to be validated in an independent cohort. However, it is the best estimate for a combined prognostic score based on our data.

Since the presence of metastases is such a strong prognostic factor, it can be discussed whether it is valid to apply a prognostic model to all sarcoma patients as a single group. Thus, we examined the prognosis in all sarcoma patients collectively as well as in the subgroup with localized disease. The SPSpib model was developed based on data from localized disease exclusively, and we found it to be a significant prognostic model with increasing HRs.

When a combined model is validated, we anticipate that this model will be easily incorporated into everyday clinical work at the centers where PET/CT scans are done as part of the initial diagnostic process in patients with sarcoma. By using a semi-automated contouring program, MTV2.5 could easily be incorporated as part of the written report for the PET/CT scan. Furthermore, a previous study has found that when evaluating a dichotomous outcome, visual assessment can stand alone [39]. However, PET/CT scans are not clinical practice according to the current guidelines [7], and the added prognostic value of MTV2.5 alone may be too small to justify such a scan.

SUVmax, MTV2.5, and TLG2.5 have previously been shown to be prognostic markers in both soft tissue and bone sarcoma, but there has been disagreement on the optimal cut-off values. We found the optimal cut-off value for SUVmax to be 16 g/mL, 56 mL for MTV2.5, and 1056 g for TLG. This is in accordance with cut-offs in some previous studies [8,19], while others have used different cut-offs [12,13,16,17,18,19,20]. This difference can be explained by different SUV cut-offs in calculating MTV, inclusion of different patient groups and types of sarcomas, different scan protocols, and different endpoints (progression-free survival, disease-free survival, metastasis-free survival, or OS). Sarcoma is a heterogeneous disease, and variation in FDG uptake between different sarcomas and within specific sub-groups is well described [9], as is also visualized in Figure 2a. As MTV2.5 could identify all tumors except one, the variation in FDG uptake explains the many outliers in Figure 2d reflecting the metabolic tumor activity and disease burden.

We tested if the MTV2.5 56 mL cut-off could predict the presence of metastases at diagnosis and found that only 2 out of 64 patients with MTV <56 mL had metastatic disease. These two tumors were atypical with either a large tumor volume with low metabolic activity or only found in a few lymph nodes without an identifiable primary tumor. On these grounds, we argue that the 56 mL cut-off can be used both in a prognostic setting and to guide the nuclear physician in detecting metastases since almost all patients with the disseminated disease were found in the group with high MTV. However, visual inspection of FDG PET/CT scans already has a high sensitivity for detecting metastases, limiting the value of added prognostic information from SUVmax, MTV, and TLG measurements.

When looking at the subgroup of patients with localized disease, we found that 4 out of 118 were inoperable. In three patients, it was technically impossible to operate due to tumor localization, and in one patient, the general condition was too poor. Among the patients with local disease at diagnosis, 20% had local recurrence, while 30% experienced metastatic recurrence in our study. This is higher than reported by others, which is possibly due to differences in included sarcoma subgroups [25,33,40]. It could have a major impact on surveillance and treatment protocols if it was possible to predict which patients were at high risk of relapse. For the patients that were operable at diagnosis, we found that SUVmax, MTV2.5, and TLG2.5 were statistically significant in predicting which of these would later experience relapse. Even though the results were significant, it is questionable if these parameters can be used in a clinical setting since the proportion of patients that experienced relapses despite a low SUVmax, MTV2.5, or TLG2.5 was high (28/72, 23/63, and 38/92, respectively). We examined the current and combined predictive models’ ability to predict relapse in localized sarcoma. With the current models, we found that patients with an HS-mGPS score of two had a significantly higher probability of experiencing relapse; this is in concordance with the original study [33]. The Choi model could not predict relapse, which was also concordant with the original study [25]. ACBS could not predict relapse, and for mACBS, only the highest score reached significance; none of these were tested for the prediction of relapse in the original studies [34,35,36]. The new models, including MTV, resulted in a significant prediction of relapse for the highest scores in MTV_HS-mGPS, MTV_ACBS, and MTV_mACBS. Comparable to the combined scores, only the highest score in the new SBSpib model significantly predicted relapse.

During the inclusion period, 435 patients had a sarcoma protocol PET/CT scan. Of these, 179 were scanned post-therapeutic. We looked closer into this group to examine if this could cause bias in the results. We found that the group was made up of patients in the flowing categories: (1) a primary sarcoma diagnosis and PET/CT was conducted prior to 2016, (2) patients with low grade sarcoma, and (3) no sarcoma suspicion upon first surgical resection. Based on these findings, it is unlikely that this subgroup would introduce significant bias in the results.

This study has several limitations. We included relatively few patients and, consequently, a small number of patients in each sarcoma subgroup. The number of patients with metastatic disease was too low for any meaningful survival analysis to be carried out on this subgroup alone. Furthermore, the follow-up period was short (mean 2.8 years), which is shorter than seen in the other studies examining prognostic scores for sarcoma [25,33,34,35,36]. It would have been preferable to examine the five-year survival rate, for example. However, a longer follow-up period was challenged by evolvement in PET/CT scanners and scanning technics. In the study population, survival was comparable among patients with STS and BS, as observed when analyzing patients with both local and metastatic disease individually. It is expected that patients with BS would have superior survival outcomes. The reason for this discrepancy in expected survival is unclear. However, it allowed us to examine them as one group. A major limitation is the retrospective nature of this study since not all patients had all biomarkers examined at diagnosis. Because of this, ALT and ESR were eliminated prior to creating the SBSpib model. Furthermore, only patients with no missing values were included, limiting the backward stepwise analysis to 65 patients. This renders the model subject to uncertainty, and external validation is needed for the results to be reliable. Unfortunately, we did not have access to a validation cohort, and we therefore highly recommend that the SBSpib model should be tested in an independent cohort, preferably where all the data is collected prospectively to avoid missing biomarker values.

## 5. Conclusions

Implementing PET parameters in current prognostic models clearly improves the prognostic value. The SBSpib model is a new combined score including both biomarkers and PET parameters, which can be used to predict OS and, to some extent, the risk of recurrence. However, the model must be validated before implementation in a clinical setting.

## Figures and Tables

**Figure 1 cancers-15-00816-f001:**
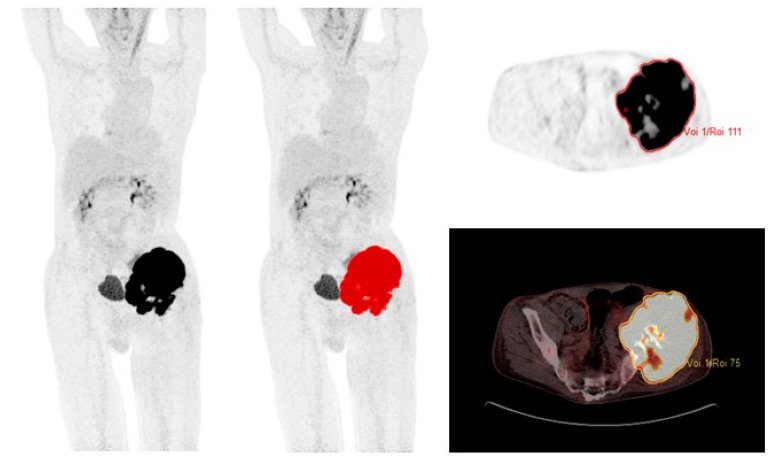
69-year-old male with localized chondrosarcoma and a volume of interest (VOI) covering the entire tumor with SUV >2.5 g/mL.

**Figure 2 cancers-15-00816-f002:**
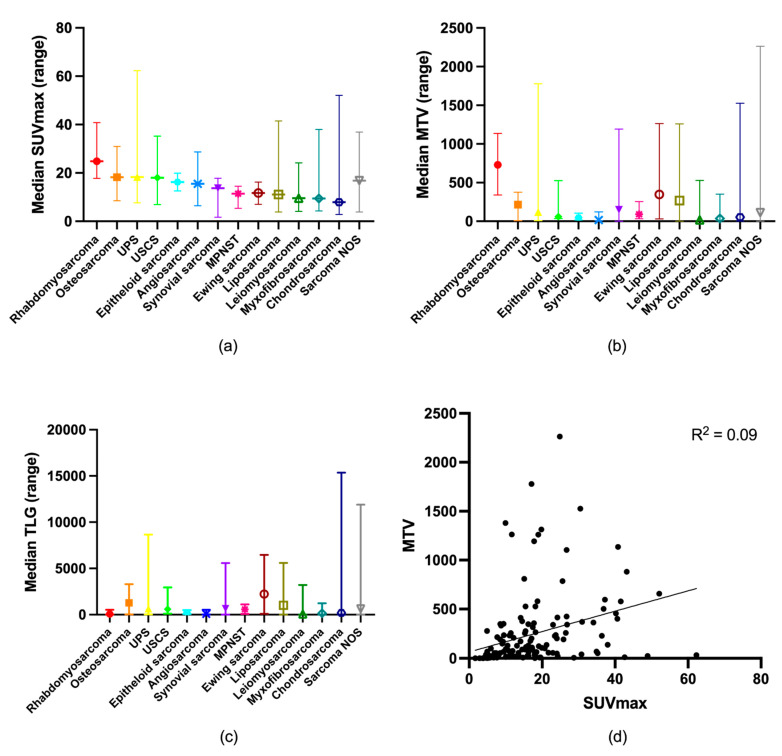
Distribution of (**a**) SUVmax; (**b**) MTV2.5; and (**c**) TLG2.5 according to sarcoma type. (**d**) Correlation between SUVmax and MTV2.5.

**Figure 3 cancers-15-00816-f003:**
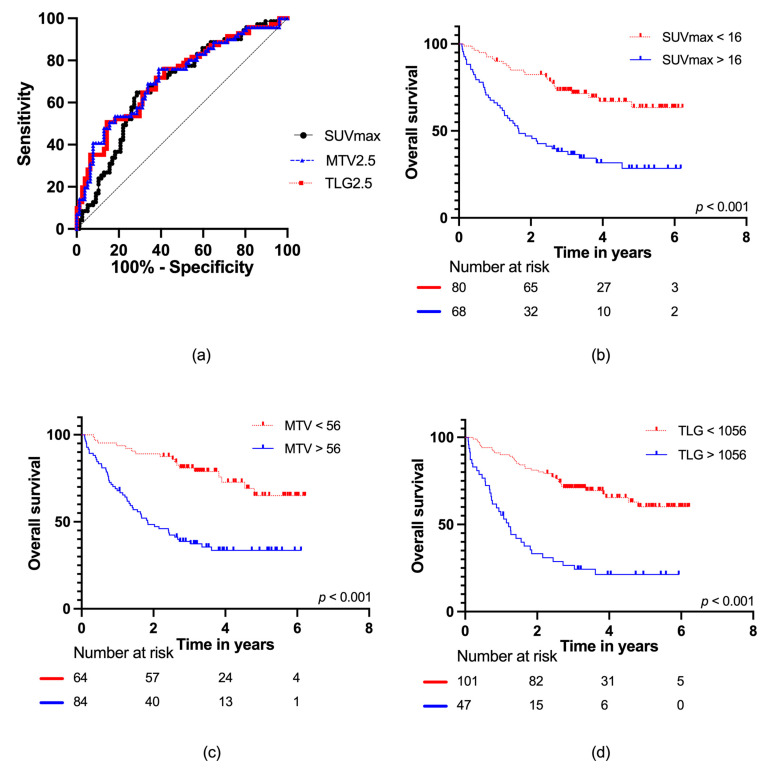
(**a**) ROC curves for SUVmax, MTV2.5, and TLG2.5 for all patients; Kaplan–Meier survival curves with number at risk at baseline, two, four, and six years for (**b**) SUVmax; (**c**) MTV2.5; and (**d**) TLG2.5.

**Figure 4 cancers-15-00816-f004:**
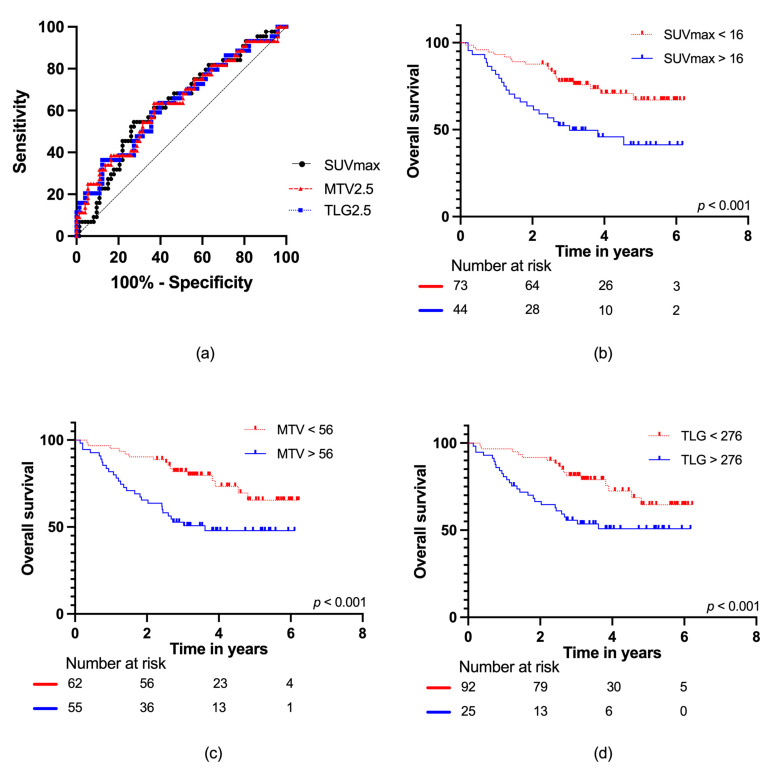
(**a**) ROC curves for SUVmax, MTV2.5, and TLG2.5 in localized disease; Kaplan–Meier survival curves with number at risk at baseline, two, four, and six years for (**b**) SUVmax; (**c**) MTV2.5; and (**d**) TLG2.5.

**Figure 5 cancers-15-00816-f005:**
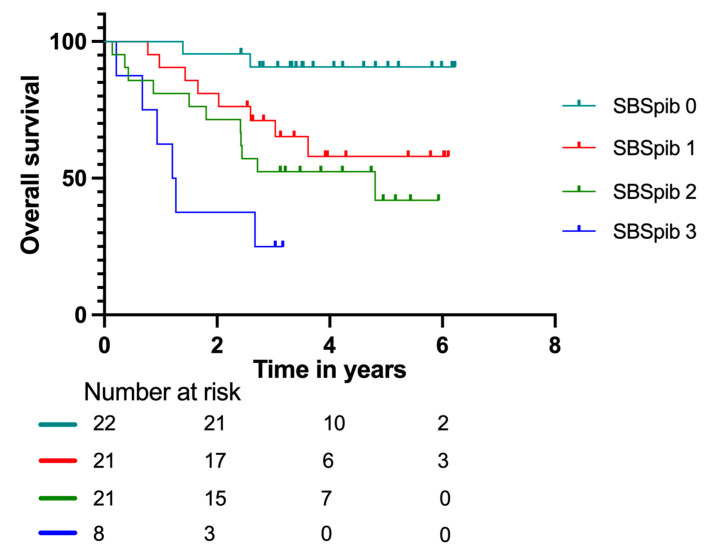
Kaplan–Meier survival curves with number at risk at baseline, two, four, and six years according to the Sarcoma Biomarker Score pet imaging and blood (SBSpib) model in patients with localized disease.

**Table 1 cancers-15-00816-t001:** Patient characteristics.

Patient Characteristic	
Male/female: *n* (%)	78 (53%)/70 (47%)
Age (years): mean (range)	61 (18–92)
**Biomarkers**	
Albumin (g/L)	
mean (range) normal/low: *n* (%) missing values: *n* (%)	37 (19–52) 71 (63%)/42 (37%) 35 (24%)
Hemoglobin (g/L)	
mean (range) normal/low: *n* (%) missing values: *n* (%)	8.3 (3.4–10.3) 93 (76%)/30 (24%) 25 (17%)
LDH (U/I)	
median (range) normal/high: *n* (%) missing values: *n* (%)	200.8 (86.9–512.7) 75 (81%)/18 (19%) 55 (37%)
ALT (U/I)	
median (range) normal/high: *n* (%) missing values: *n* (%)	20.3 (8.7–88.7) 53 (91%)/5 (9%) 90 (61%)
ESR (mm)	
median (range) normal/high: *n* (%) missing values: *n* (%)	17 (2–113) 28 (55%)/23 (45%) 97 (66%)
CRP (mg/L)	
median (range) normal/high: *n* (%) missing values: *n* (%)	10 (1–296) 51 (50%)/51 (50%) 46 (31%)
Neutrophils (10^9^/L)	
median (range) normal/high: *n* (%) missing values: *n* (%)	5.6 (2.3–27.8) 80 (76%)/25 (24%) 43 (29%)
Lymphocytes (10^9^/L)	
mean (range) normal/low: *n* (%) missing values: *n* (%)	1.7 (0.4–4.2) 70 (69%)/32 (31%) 46 (31%)
Sodium (mmol/L)	
mean (range) normal/low: *n* (%) missing values: *n* (%)	140 (130–147) 107 (86%)/17 (14%) 24 (16%)
**Tumor characteristic**	
Localised/disseminated: *n* (%)	117 (79%)/31 (21%)
Angiosarcoma: *n* (%)	8 (5.4%)
Epithelioid sarcoma: *n* (%)	4 (2.7%)
Ewing sarcoma: *n* (%)	<4 (<2.7%)
Chondrosarcoma: *n* (%)	28 (18.9%)
Leiomyosarcoma: *n* (%)	12 (8.1%)
Liposarcoma: *n* (%)	14 (9.5%)
Malignant peripheral nerve sheath tumor (MPNST): *n* (%)	<4 (<2.7%)
Myxofibrosarcoma: *n* (%)	11 (7.4%)
Osteosarcoma: *n* (%)	8 (5.4%)
Undifferentiated pleomorphic sarcoma (UPS): *n* (%)	24 (16.2%)
Rhabdomyosarcoma: *n* (%)	4 (2.7%)
Synovial sarcoma: *n* (%)	6 (4.1%)
Undifferentiated spindle cell sarcoma (USCS): *n* (%)	14 (9.5%)
Sarcoma NOS: *n* (%)	9 (6.1%)
STS total: *n* (%)	109 (74%)
BS total: *n* (%)	39 (26%)
Total: *n* (%)	148 (100%)
**PET parameters**	
SUVmax (g/mL): median (range)	13.53 (1.68–62.29)
MTV2.5 (mL): median (range)	73.89 (0–2264.06)
TLG2.5 (g): median (range)	351.45 (0–15,364.82)

**Table 2 cancers-15-00816-t002:** Model testing.

Score	All Patients			Localized Disease		
	*n*	AIC	C-Index	*n*	AIC	C-Index
MTV2.5	95	376	0.65	72	208	0.65
HS-mGPS	95	390	0.64	72	216	0.62
MTV_HS-mGPS	95	377	0.71	72	211	0.68
MTV2.5	51	185	0.60	43	127	0.58
Choi	51	188	0.59	43	127	0.58
MTV_Choi	51	184	0.66	43	127	0.63
MTV2.5	94	368	0.66	71	200	0.65
ACBS	94	376	0.67	71	204	0.64
MTV_ACBS	94	365	0.72	71	201	0.70
MTV2.5	94	368	0.66	71	200	0.70
ACBSm	94	377	0.68	71	204	0.66
MTV_ACBSm	94	367	0.73	71	202	0.65

**Table 3 cancers-15-00816-t003:** Hazard ratios for overall survival in combined predictive models.

Score	All Patients				Localized Disease			
	*n* (%)	Events	HR (95% CI)	*p*-Value	*n* (%)	Events	HR (95% CI)	*p*-Value
MTV_HS-mGPS								
0	25 (26)	6	1		24 (33)	5	1	
1	34 (36)	14	1.88 (0.72–4.90)	0.20	26 (36)	8	1.51 (0.49–4.62)	0.47
2	11 (12)	9	5.35 (1.89–15.12)	<0.01 *	6 (8)	5	5.86 (1.69–20.37)	<0.01 *
3	25 (26)	18	5.40 (2.13–13.66)	<0.01 *	16 (22)	9	3.79 (1.27–11.34)	0.02 *
Total	95 (100)	47			72 (100)	27		
MTV_Choi								
0	8 (16)	3	1		7 (16)	2	1	
1	30 (59)	12	1.16 (0.33–4.11)	0.82	26 (61)	8	1.05 (0.22–4.94)	0.95
2	13 (25)	11	4.76 (1.31–17.26)	0.02 *	10 (23)	8	4.92 (1.04–23.37)	0.045 *
Total	51 (100)	26			43 (100)	18		
MTV_ACBS								
0	16 (17)	2	1		16 (22)	2	1	
1	24 (26)	8	2.89 (0.61–13.60)	0.18	19 (27)	5	2.12 (0.41–10.95)	0.37
2	17 (18)	9	5.53 (1.19–25.63)	0.03 *	12 (17)	4	3.00 (0.55–16.42)	0.20
3	37 (39)	27	10.40 (2.47–43.84)	<0.01 *	24 (34)	15	7.16 (1.63–31.33)	<0.01 *
Total	94 (100)	46			71 (100)	26		
MTV_ACBSm								
0	16 (17)	2	1		16 (23)	2	1	
1	24 (25)	8	2.89 (0.61–13.59)	0.22	19 (27)	5	2.12 (0.41–10.95)	0.37
2	16 (17)	9	6.00 (1.29–27.83)	0.02 *	11 (15)	4	3.31 (0.61–18.10)	0.17
3	11 (12)	8	8.15 (1.73–38.44)	<0.01 *	8 (11)	5	5.75 (1.12–29.68)	0.04 *
4	27 (29)	19	10.90 (2.53–46.95)	<0.01 *	17 (24)	10	7.40 (1.62–33.81)	0.01 *
Total	94 (100)	46			71 (100)	26		

* indicates statistical significance.

**Table 4 cancers-15-00816-t004:** Hazard ratios for overall survival in the new model.

SBSpib in Localized Disease				
Score	*n* (%)	Events	HR (95% CI)	*p*-Value
0	22 (31)	2	1	
1	21 (29)	8	4.83 (1.02–22.75)	0.047 *
2	21 (29)	11	7.40 (1.64–33.42)	0.01 *
3	8 (11)	6	17.32 (3.45–86.93)	<0.01 *
Total	72 (100)	27		

* indicates statistical significance.

## Data Availability

The data that support the findings of this study are available from the corresponding author upon reasonable request.

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
