# Peer review of "Inclusion of Metabolic Tumor Volume in Prognostic Models of Bone and Soft Tissue Sarcoma Increases the Prognostic Value"

_cancers, 2023, doi:10.3390/cancers15030816_

Round 1

Reviewer 1 Report (Previous Reviewer 1)

The authors improved the manuscript significantly. All comments have been taken into account. 

Author Response

The comment is highly appreciated.

Reviewer 2 Report (Previous Reviewer 2)

Thank you for the corrections.

However, as I do not have any response letter from authors after corrections, in lines 271-273 it is not clear what is ment with 25/80, 46/68; 17/64 vs 54/84 and 35/101 vs. 36/47.  If these are patients (as it seems to be in the next page), a percentage is enough information, as this way is not clear to reader. The patient numbers are more useful in the figure 3.

The same phenomen is in the lines 305-306. Please add the word patient or remove 20/73 etc or add these numbers to the figure 4, where they are most useful.

The patient numbers would be useful also for figures S1.

Author Response

We very much appreciate the comments and revisions have been made accordingly:

Section 3.4: Number of patients have been deleted

Figure 3: Number at risk have been added

Figure 4: Number at risk have been added

Figure 5: Number at risk have been added

Figure S1: Number at risk have been added

Reviewer 3 Report (Previous Reviewer 3)

The manuscript has been significantly improved and now I support its publication in Cancers.

Author Response

The comment is highly appreciated.

This manuscript is a resubmission of an earlier submission. The following is a list of the peer review reports and author responses from that submission.

Round 1

Reviewer 1 Report

It is a well written manuscript. However, I believe there are some major limitations/concerns to this study that should be taken into account:

-   The linear predictors were categorized. I would suggest including all the biomarkers and linear PET parameters as continuous variables and if needed add non-linear terms. This would be more informative and could improve performance.

-    Table 1should also include the number of missing values for each parameter. In the discussion it was stated that not all biomarkers were examined. These biomarkers were recorded as normal for this study. I believe that using multiple imputation techniques to impute these missing values would result in more reliable results.

-    What is the C-statistic of the new model? Could you add this to paragraph 3.8?

-     What are the C-statistics of the ROC curves of figure 3a and figure 4a.

-     You assessed the discriminative ability of the updated models, however what is the calibration of the models (observed vs predicted probability)?

-     Score 4 of the new model results in a hazard ratio of 650. This group included 4 patients with a number of events of 4. This result is therefore not reliable. Also the other subgroups are relatively small. I think that without external validation of the results, the added value of these results is not assessable.

- PET/CT is not clinical practice according to the current clinical guidelines (ESMO/NCCN) (should only be used as problem-solving tool in selected cases). I believe that the added value of MTV2.5 is too small to justify a PET/CT in clinical practice. 

Reviewer 2 Report

Thank you for authors of this interesting paper. However, some questions are suggested to discuss/correct.

In line 54, prognostic markers are listed. How about surgical margin and tumor depth?

Are the blood values routinely taken in your hospital? Were there many patients excluded because of missing values?

How were patients selected to the PET/CT, as in the study years it was not a standard procedure everywhere in Europe? Most patients were excluded because of missing pretreatment PET/CT, can this cause bias, especially consider the paragraph 3.6? How many patients did not have PET/CT at all?

Please write open VOI in the Figure 1.

Is the VOI same also in MTV_Choi (line 160)? It is not mentioned there, although it is mentioned in every other prognostic score.

Line 210: please change Table I->Table 1.

In table 1 it would be more informative if patients with normal and abnormal values would be presented as percentages. These should be removed from text to table (line 228 forwards).

You should discuss why bone and soft tissue sarcomas had the same OS and were grouped. Usually STSs have a lot worse survival and these two are very different sarcoma entities.

Some patients had very short follow-up time. When this data was collected and would the results be different with longer follow-up time? Please discuss. What were the patient numbers and follow-up time in other prognostic scores mentioned in the paper?

I also think that inoperable and operable malignancies and on the other hand local and metastasized sarcomas cannot be in the same prognostic score, they differ in prognosis very much. This is clear in every other malignancy. Please justify or change.

The sentences beginning in the line 248, 261 and 273 are unclear, please clarify.

Why the number of patients is marked with <4 in many places? Please indicate the real number or be more clear otherwise.

Please explain why patient numbers in Table 3 differ. This should be mentioned also in Methods paragraph.

Line 360: Do you have a reference for the statement that PET/CT should be done in every sarcoma patient, as this is not a standard everywhere?

The information in the line 412 should be in Methods -section. There also should be stated how many missing values were recorded as normal when not known an actual value.

So actually you have only 22 patients with valid information.  This should be evaluated more, what does it mean to the whole study? Bias, percentage of guessed values, etc.

Please tell also, who did the measurements? Oncologist or do you have a protocol so all this information is available oncologist? How easy this would in clinical use?

Reviewer 3 Report

In this paper, Pedersen et al. study the addition of metabolic tumor volume in prognostic models of bone and soft tissue sarcoma and find that this increases the prognostic value.

General comment: The topic of the study is relevant and important. The paper is well written.

Specific points:

Section 2.5: Would it be possible to present the New prognostic scores in a Table for easier inspection?

Kaplan-Meier survival curves are missing p-values.

It is not clearly stated what are the results of a multivarate analysis.
